# Polar and Non-Polar Zn_1−x_Mg_x_O:Sb Grown by MBE

**DOI:** 10.3390/ma15238409

**Published:** 2022-11-25

**Authors:** Ewa Przezdziecka, Karolina M Paradowska, Rafal Jakiela, Serhii Kryvyi, Eunika Zielony, Ewa Placzek-Popko, Wojciech Lisowski, Piotr Sybilski, Dawid Jarosz, Abinash Adhikari, Marcin Stachowicz, Adrian Kozanecki

**Affiliations:** 1Institute of Physics, Polish Academy of Sciences, Al. Lotników 32/46, 02-668 Warsaw, Poland; 2Łukasiewicz Research Network—Institute of Microelectronics and Photonics, Al. Lotników 32/46, 02-668 Warsaw, Poland; 3Department of Quantum Technologies, Wroclaw University of Science and Technology, Wybrzeze Wyspianskiego 27, 50-370 Wroclaw, Poland; 4Institute of Physical Chemistry, Polish Academy of Science, 01-224 Warsaw, Poland; 5Institute of Materials Engineering, Center for Microelectronics and Nanotechnology, University of Rzeszow, ul. Pigonia 1, 35-959 Rzeszow, Poland; 6International Research Centre MagTop, Institute of Physics, Polish Academy of Sciences, 02-668 Warsaw, Poland

**Keywords:** ZnO, MBE, Raman spectroscopy, secondary ion mass spectroscopy, optical properties

## Abstract

The article presents a systematic study of Sb-doped Zn_1−x_Mg_x_O layers, with various concentrations of Mg, that were successfully grown by plasma-assisted MBE on polar *a*- and *c*-oriented and non-polar *r*-oriented sapphire substrates. X-ray diffraction confirmed the polar *c*-orientation of alloys grown on *c*-and *a*-oriented sapphire and non-polar structures grown on *r*-oriented substrates. A uniform depth distribution of the Sb dopant at level of 2 × 10^20^ cm^−3^ was determined by SIMS measurements. Raman spectroscopy revealed the presence of Sb-related modes in all samples. It also showed that Mg alloying reduces the compressive strain associated with Sb doping in ZnO. XPS analysis indicates that the chemical state of Sb atoms in ZnMgO is 3+, suggesting a substitutional position of Sb_Zn_, probably associated with two V_Zn_ vacancies. Luminescence and transmission spectra were measured to determine the band gaps of the Zn_1−x_Mg_x_O layers. The band gap energies extracted from the transmittance measurements differ slightly for the *a*, *c*, and *r* substrate orientations, and the differences increase with increasing Mg content, despite identical growth conditions. The differences between the energy gaps, determined from transmission and PL peaks, are closely correlated with the Stokes shift and increase with the Mg content in the analyzed series of ZnMgO layers.

## 1. Introduction

The band gap of ZnO can be widened without changing its crystal structure by incorporating isovalent Mg into the ZnO matrix. Zn_1−x_Mg_x_O ternary alloy is considered the most suitable barrier layer for carrier confinement due to its similar lattice constant to that of ZnO, and its tunable wide band gap (in the range from 3.37 eV to about 4 eV, according to its composition x), and thus it is a good material for ZnO/Zn_1−x_Mg_x_O superlattices and quantum wells [1,2,3,4,5]. ZnO and its alloy with MgO have extensive application prospects in the area of short-wavelength optoelectronic devices. Moreover, as it was shown, Zn_1−x_Mg_x_O alloys can be successfully p-type doped. The p-Zn_1−x_Mg_x_O films were previously realized with phosphors (P) [6], lithium (Li) [7], antimony (Sb) [8] mono-doping, indium plus nitrogen (In–N) [9], and boron plus phosphorus (B-P) [10] co-doping methods. Despite many published papers on such doping, some obstacles toward practical ZnO-bases devices remained. In the case of Sb doping, a p-type ZnO:Sb film [11,12], as well as n-type [13] layers, were obtained. In the case of doping with Sb, it has been found that these dopants have a low solubility when replacing oxygen in a ZnO matrix, due to the large ionic radii mismatch [13]. This fact is also supported by theoretical calculations, which suggest that these elements should become more stable when localized on zinc sites. A model for a large-sized mismatched group V dopant, such as Sb in ZnO, was reported, in which a Sb_Zn_-2V_Zn_ complex was thought to be a suitable acceptor instead of a deep Sb_O_ acceptor [14]. Thus, gaining deeper knowledge on the acceptor doping in Zn_1−x_Mg_x_O is strongly needed.

Usually, most ZnO-based devices and structures are grown along the polar c-axis direction. However, polar heterostructures experience strong spontaneous and piezoelectric polarization fields, which deteriorate the internal quantum efficiency of the light emitting diodes (LEDs). The exploration of non-polar ZnO based films and quantum structure growth to avoid the polarization effects are ongoing [15]. On the other hand it was also proven that the polarity of the oxide films affects the doping efficiency of ZnO [16]. Therefore, the fundamental studies on acceptor doping in Zn_1−x_Mg_x_O thin films are crucial towards the fabrication of ZnO based devices. The direct comparison of optical, structural, and other properties of polar and non-polar Sb doped Zn_1−x_Mg_x_O layers grown in the same growth conditions and with the same growth method was not studied sufficiently so far. In this paper the structural and optical properties of Zn_1−x_Mg_x_O:Sb polar and nonpolar films are systematically studied and compared.

## 2. Materials and Methods

Zinc magnesium oxide layers (Zn_1−x_Mg_x_O) doped with antimony (Sb) were grown on commercially available *r*-, *c*-, *a*-oriented Al_2_O_3_ substrates by plasma assisted molecular beam epitaxy (PA-MBE) using a Riber Compact 21 system. In one process, series of samples on differently oriented substrates were grown. As a result, with the same growth parameters, Zn_1−x_Mg_x_O:Sb layers with different preferential orientations were obtained. Knudsen cells were used as the sources of antimony and zinc, while radiofrequency (rf) plasma cells were used as a source of oxygen. During the growth, the rf power of the O_2_ source was ~400 W and it had a 3 mL/min flow. The Sb Knudsen cell tip temperature was fixed at 460 °C and the growth temperature during all processes was the same and it was about 450 °C. The thickness of the Zn_1−x_Mg_x_O:Sb layers was ~0.4–0.5 µm.

The compositional depth profiles and concentration of atoms were determined by secondary ion mass spectrometry (SIMS) using ion implanted standards as references. Temperature dependent photoluminescence (PL) was measured within the 10–300 K range. The 302.5 line of an Ar ion laser was used as the excitation source. RT optical transmittance spectra were studied by a Varian Cary 5000 spectrophotometer Agilent in the range from 250 to 500 nm at room temperature (RT). Structural X-ray diffraction measurements were carried out with the Cu K_α1_ radiation using a Bragg−Brentano powder diffractometer (X’Pert Pro Alpha1 MPD from Philips/PANalytical) equipped with an incident beam Ge(111) Johansson monochromator and a strip detector.

The electronic properties of Sb were studied with X-ray photoemission spectroscopy (XPS). The XPS spectra were recorded with a PHI 5000 VersaProbe™ scanning Physical Electronics ESCA Microprobe using monochromatic Al-Kα radiation (hν = 1486.6 eV) from an X-ray source operating with 100 μm spot size, 25 W power, and 15 kV acceleration voltage. The binding energy (BE) scale refers to the Fermi level and was adjusted to the carbon XPS peak position (C 1s-at the binding energy of 284.8 eV).

Room temperature Raman spectra were measured in the range of 70–1200 cm^−1^ with the use of integrated T64000 Jobin-Yvon system LabSpec 5 for Raman- and micro-Raman measurements at a single mode operation with a liquid nitrogen cooled high-resolution Si charge-coupled device (CCD) and an Ar laser (514.5 nm) as a light source. The usual laser power during the measurement was 20 mW and the light spot had the diameter of 2 µm. Samples were measured in the backscattering geometry from the grown surface (c- or a-plane) either with or without polarization detection, depending on the investigated sample. Polar samples were measured without polarization detection, which corresponds to $z\left(-,-\right)\bar{z}$ in Porto notation. For nonpolar samples, the Raman spectra were varying depending on the sample orientation under the microscope, so we measured each sample on the rotational table, collecting the spectra in the rotation range of 0–180° with the step of 10° and using the analyzer on scattered light and the natural polarization of the laser light. Then, based on the ZnO modes intensity for each sample, one spectrum was chosen for further analysis that corresponded to $y\left(x,x\right)\bar{y}$ geometry in Porto notation.

## 3. Results

### 3.1. Samples and SIMS Measurements

Series of Zn_1−x_Mg_x_O:Sb layers were grown on *r*-, *c*-, *a*-oriented Al_2_O_3_ substrates. Numbers 1, 2, 3, and 4 indicate different Mg concentrations in layers changed by the Mg cell temperature (Table 1), and letters A, C, and R next to the sample numbers used in text and figures indicate series grown on *a*-, *c*-, *r*-oriented Al_2_O_3_ substrates, respectively. The concentration of Mg was changed by controlling the tip and base temperature of the Mg effusion cell. In order to check the Mg and Sb content in all samples, the SIMS measurements were performed. A cesium (Cs+) primary beam at the energy of 5.5 keV with the current kept at 50 nA was used. The size of the raster is about 150 × 150 µm^2^ and the secondary ions are collected from a central region of 60 µm in diameter. Secondary ions ^16^O^133^Cs+, ^24^Mg^133^Cs+, and ^121^Sb^133^Cs+ were collected by the electron multiplier. As it is shown in Figure 1a, the concentration of Mg atoms in the series of Zn_1−x_Mg_x_O layers changes as expected (blue lines represent the Mg signal), whereas the concentration of Sb atoms stays at the same level of 2 × 10^20^ at./cm^3^ in all samples (red lines represent the Sb concentration obtained based on ion implanted standards for ZnO). It is worth noting that the concentration of Sb in the same set of samples (with the same concentration of Mg and grown with the same process, but on differently oriented substrates) does not depend on the used substrate.

### 3.2. Structural Analysis

The crystal structures of the films were determined by x-ray diffraction (XRD). The XRD patterns show high intensity ZnO 0002 and 0004 peaks for samples grown on *a*- and *c*-oriented Al_2_O_3_ substrates, indicating that Zn_1−x_Mg_x_O:Sb thin films have a wurtzite structure and a preferred orientation along the *c* axis (Figure 2a,b). Based on simple analysis (assuming that the entire XRD signal from the layer is visible in the theta-2theta scans and keeping in mind the peaks intensity relationship in fully polycrystalline ZnO powders), we observe about 99–98% of *c*-oriented Zn_1−x_Mg_x_O for the samples grown on *c*- and *a*-sapphire, except for those with a relatively high concentration of Mg. For the *c*-oriented Zn_1−x_Mg_x_O, the percentage is about 91% on *c*-sapphire and 96% on *a*-sapphire. In these cases, the 10–11 orientation of a layer is also observed with 9% or 4%, respectively. For sample 4C with the highest concentration of Mg, X-rays reflected from the second plane of atoms (20–22) is also observed. The intensity of the X-rays reflected from the first plane of atoms (10–11) is in this case intensive, which makes the observation of second planes possible.

Let us focus on the position and the FWHM (full-width at half maximum) of the peaks 0002 and 0004 as a function of Mg content in the ZnO material. It can be noted that both of the positions of the peaks slightly shift toward higher values of θ/2θ with increasing Mg content, and FWHM also increases with increasing Mg content. Such a behavior can be explained by a reduction in the volume of the unit cell due to the incorporation of Mg and, consequently, leads to the decrease in the *c*-lattice parameter. Additionally, the FWHM of the 0002 peak is larger for the samples grown on *c*-sapphire, as it can be seen in Figure 3.

In the case of the samples grown on *r*-sapphire, the 11–20 peak is dominant, and 99–100% of the *a*-oriented layer is detected, except for the samples with a higher concentration of Mg, where only 28% of the layers are (11–20) oriented and 72% of the contribution represents (0001) orientation (Figure 2c).

Thus, the layers of Zn_1−x_Mg_x_O grown on the *c*- and on the *a*-sapphire substrates are polar, whereas those grown on the *r*-sapphire substrates are nonpolar. The samples with a high Mg concentration exhibit polycrystalline properties. The problem with the homogeneity of Zn_1−x_Mg_x_O for higher Mg concentrations is repeatedly reported and is correlated with the fact that MgO crystallizes in a cubic structure instead of a hexagonal one [17,18]. The thermodynamic solubility limit of MgO in the wurtzite phase of the Zn_1−x_Mg_x_O alloy was reported to be x = 0.04 in a bulk form [19] and the solubility limit is much higher for epitaxial thin films. In our samples, with a high concentration of Mg, pure wurtzite phases were detected, but with a polycrystalline character with a preferential orientation. Both the *c* and *a* lattice parameters change with Mg concentration, as is presented in (Figure 4), where the lattice constant versus the MgCs+/OCs+ SIMS ion signals ratio are shown. The “a” lattice parameter is extracted from the position of 11–20 peaks on nonpolar layers, whereas “c” is from the position of 0002 peaks in the case of polar layers.

### 3.3. Optical Properties

Transmission and PL measurements were carried out in order to precisely determine the variations of the bandgap and the dominant PL transitions depending on the Mg concentration and the orientation of the substrate. Based on the analysis of the transmittance spectra, we observe that the bang gap increase duo to Mg doping (Table 2 and Figure 5a). We have also observed that the energy gaps of the layers grown under the same conditions depend to the same extent on the orientation of the substrate. These differences increase with the concentration of Mg (Figure 5b).

It is well known that the band-edge and the near band-edge region of the optical spectrum of ZnO films depend on the substrate and on postdeposition annealing. It has been reported that, in the case of pure ZnO, usually the band gap of ZnO on *c*-plane sapphire is reported to be lower than that on *r*-plane sapphire and it is consistent with our results. In our samples, the bandgap difference between ZnO deposited on *r*- and *c*-sapphire is about 49 meV and it changes with Mg concentration. It should be noted that the thermal mismatch of the lattice constants of ZnO on the *c* and *r* planes of Al_2_O_3_ goes in opposite directions and it can affect the stress in the layers [20].

As expected, the position of the PL peaks depends on the Mg concentration (Figure 6a). At low temperatures, in all samples with a small concentration of Mg, peaks related to free excitons (FX) (peak is usually visible at a temperature higher than 50 K) and donor-bound excitons (D^0^X) (~3.36 eV) are observed. We also identified an emission band located near 3.31 eV, which can be decomposed into two peaks which originate from free electrons-to-bound transitions (FA) and donor-acceptor pairs (DAP) (Figure 6b) [21]. In samples with the highest Mg content (sample 4) two peaks are still visible (Figure 6b). Due to Mg alloying, FWHM of the PL peaks increases, similarly to the dependence reported in [22]. The characteristic “s” shape is observed in the temperature dependence of peak positions of the PL in samples with a high Mg concentration (Figure 7). This behavior is typical of exciton recombination in alloys with random composition fluctuations. At low temperatures, excitons are trapped in areas with a low Mg content (local energy minima) and then, as the temperature increases, they can also fill higher composition areas, contributing to higher energy emission. We observed that the position of the PL depends on the orientation of the sapphire substrate, which may suggest either a slight difference in the composition of the alloys or the presence of stresses. The energetic position of the PL broad band measured at 10 K was 3.66 eV for sample 4A, for sample 4C it was 3.68 eV, and for sample 4R it was 3.61 eV.

The energy gap values extracted from the transmittance data measured at room temperature E_g_^RT^ were compared with the positions of the PL peaks measured at 10 K E_PL_^10K^ (Figure 8a). The energy difference between these two parameters Δ = E_g_^RT^ − E_PL_^10K^ is called a delta parameter Δ and is closely correlated with the value of Stokes shift values (unfortunately in our case, transmittance was measured at RT and PL at 10 K). Stokes shift is the difference between the positions of the absorption and emission spectra of the same electronic transition. As it is shown in Figure 8b, the Δ changes with Mg concentration in Zn_1−x_Mg_x_O layers. Moreover, it is clear that this shift depends on the orientation of the substrate. In agreement with [22], this shift increases with an increasing Mg concentration for the layers grown on *a*- and *r*-plane substrates, which suggests a more pronounced composition fluctuation in these layers than in the case of the c-plane substrates. The energetic difference between absorption and PL peak increases gradually as *x* increases. This effect may be explained by larger fluctuations of the composition, which strongly influence the local band gap values. A deepening of the potential fluctuations with increasing *x* results in the stronger localization of excitons and the increase of the Stokes shift. Similar differences can be observed in the case of FX and FA transitions [23]. However, such behavior is not observed in the case of *c*-oriented substrates and the possible reason can be related to a different stress, depending on the substrate orientation.

### 3.4. Raman Spectroscopy

The vibrational properties of investigated layers were examined with the use of Raman spectroscopy. The room temperature Raman spectra were measured either with no polarization detection in $z\left(-,-\right)\bar{z}$ in Porto notation (*c*-Zn_1−x_Mg_x_O) or with polarization detection on the rotational table, which allowed for the extraction of the spectra measured in $y\left(x,x\right)\bar{y}$ configuration according to Porto notation (*a*-Zn_1−x_Mg_x_O). Collected spectra are presented in Figure 9a together with the spectra of corresponding substrates. The spectra contain a number of Al_2_O_3_ modes at 378, 418, 431, 450, 577, 645, and 751 cm^−1^, marked above the corresponding lines in substrate spectra with their origin [24]. The spectra also contain both ZnO $E_{2}^{high}$ and $E_{2}^{low}$ modes (Figure 9b). Moreover, additional modes appear in some spectra–a broad band between 500 and 600 cm^−1^, a 240 cm^−1^ line and 333 cm^−1^ line. Both the broad band at 500–600 cm^−1^ and a 240 cm^−1^ line are ascribed to the presence of Sb dopant; the 500–600 cm^−1^ band was previously discussed by us in another paper [25], while the 240 cm^−1^ line is mentioned in few papers as a local vibrational mode related to Sb occupying the Zn site of the ZnO host lattice [26,27,28]. The 333 cm^−1^ line is a second-order ZnO E_2_^high^–E_2_^low^ mode [29]. Sharp lines between 70 and 150 cm^−1^ are artefacts appearing in all spectra measured with this measurement setup.

It can be noted that spectra of *c*-Zn_1−x_Mg_x_O:Sb/*a*-Al_2_O_3_ samples exhibit the most features associated with Sb doping–the 500–600 cm^−1^ band is clearly visible as well as the 240 cm^−1^ line. The *c*-Zn_1−x_Mg_x_O:Sb/*c*-Al_2_O_3_ layers on the other hand exhibit the most intense ZnO-related lines, the second-order E_2_^high^-E_2_^low^ line appears in the spectra of all investigated samples, while for *c*-Zn_1−x_Mg_x_O:Sb/*a*-Al_2_O_3_ it appeared only in the sample without Sb. The spectra of *a*-Zn_1−x_Mg_x_O:Sb/*r*-Al_2_O_3_ layers do not exhibit easy recognizable ZnO-related modes, which may indicate the lowest crystal quality of the Zn_1−x_Mg_x_O films.

In order to analyze positions and FWHM of ZnO E_2_ modes, part of the spectra between 420 and 460 cm^−1^ was fitted with Lorentz functions.

The analysis of E_2_ modes yields several patterns–the summary of positions and FWHM of Raman E_2_ modes, obtained from fitting data with Lorentz functions, are presented in Table 2. First of all, it was not possible to analyze all spectra because, for higher Mg contents, the E_2_ modes almost vanished and were almost invisible due to the artefacts surrounding the E_2_^low^ mode and the Al_2_O_3_ E_g_ modes surrounding the E_2_^high^ mode. The E_2_ modes are a “fingerprint” of the ZnO wurtzite structure, so this may indicate that the Mg concentration was so high that the wurtzite structure deteriorated enough to cause the disappearance of E_2_ modes. This happened in the case of E_2_^low^ mode for all samples 4 and for samples 3 grown on *a*- and *r*-Al_2_O_3_, as well as for E_2_^high^ mode of sample 4 grown on *r*-Al_2_O_3_. Second of all, sample 1 always exhibit narrower E_2_ lines, regardless of the substrate, which is a result of the lack of Sb dopant in this sample. Third of all, the FWHM of E_2_ modes generally increase with Mg content, with one exception being sample 4A grown on *a*-Al_2_O_3_ substrate. However, when comparing this result with those shown in Figure 9b, one will notice that the mode was almost invisible and very close to the Al_2_O_3_ E_g_ mode at 430 cm^−1^ that greatly affected the spectrum in this range; therefore it may be just a result of a poor fitting process. The increase in FWHM indicates the disorder in the sample, so the fact that samples with Sb or higher values of Mg content exhibit wider Raman lines is normal and expectable. The peak broadening becomes greater for the sample with a higher Mg content (Figure 9) and Mg deforms the ZnO lattice, but no MgO related modes are visible, suggesting that Mg has very similar coordination to Zn.

The analysis of the positions of E_2_ modes is more complicated because there are few things that can affect the positions of Raman modes. The shift usually indicates the presence of the strain in the layers, but the origin of the strain may vary. Here, we have to take three factors into account: the substrate, the Sb dopant, and the Mg addition. In the strain-free crystal, the E_2_^low^ and E_2_^high^ modes appear at 99 and 437 cm^−1^, respectively. The data in Table 3 shows that the E_2_^high^ mode usually shifts towards higher values of the Raman shift, which indicates the presence of compressive strain; the E_2_^low^ line however is impossible to analyze due to the lack of data.

Previously, we studied *c*-ZnO:Sb layers grown on *a*-Al_2_O_3_ substrates and the investigation showed that the E_2_^high^ mode shifted towards higher values of the Raman shift with increasing Sb content, which was correlated with biaxial in-plane stress in the oxygen sublattice due to antimony ions (7–10%) occupying oxygen sites [30]. When analyzing the data for Zn_1−x_Mg_x_O:Sb samples in terms of the used substrate, it can be noted that the shift of the E_2_^high^ mode is generally larger for samples grown on *r*-Al_2_O_3_ than for any other substrate, which may indicate that ZnO grown on *r*-Al_2_O_3_ is more strained than when grown on *a*- or *c*-Al_2_O_3_. One may also notice that, for samples grown on *r*- and *c*-Al_2_O_3_, the sample with no Sb dopant exhibits the largest E_2_^high^ shifts, while for *a*-Al_2_O_3_ this sample has the lowest shift. This may indicate that the a-oriented substrate introduces the lowest strain into the ZnMgO layers while using *r*- and *c*-oriented sapphire results in compressive strain. However, in further analysis, the nature of E_2_ modes has to be taken into consideration. The E_2_ modes represent displacements perpendicular to the *c*-axis (*z*-axis). The modes oscillate linearly independently from the *x*- and *y*-direction with the same energy which leads to the twofold degeneracy of the E_2_ modes. The E_2_^high^ mode corresponds mostly (~85%) to the vibration of oxygen atoms and is almost insensitive to the mass substitution on the cation site [31]. However, what is important is that the E_2_^high^ mode is associated with atomic motion in the *xy* plane and depends on the in-plane lattice dimension in the case of *c*-oriented ZnO and out-of-plane in the case of *a*-oriented layers, where *x* and *y* are polarization directions along the hexagonal crystallographic (1-100) and (11-20) directions [32]. The decrease of the E_2_^high^ mode frequency was previously observed in ZnO under tensile strain along the *xy* plane [33].

When taking into account the Sb dopant, one may notice that, for the samples grown on *r*- and *c*-Al_2_O_3_, the largest E_2_^high^ shifts are observed in the spectra of the samples without Sb dopant. In the case of undoped but grown on *a*-Al_2_O_3_ samples, the spectra exhibit the lowest shift of E_2_^high^.

When analyzing the data in terms of MgCs+/OCs+ SIMS ion signals ratio, the conclusion is that the E_2_^high^ shift decreases for samples with a higher Mg content (Figure 10). This could indicate that incorporating Mg into ZnO introduces tensile strain which counteracts the strain introduced by the Sb dopant or chosen substrate. For *c*-Al_2_O_3_, the shift seems to be the smallest and, for the sample with the highest Mg content, the shift changes the sign and the E_2_^high^ becomes slightly shifted towards lower values, indicating tensile strain.

The E_2_^Low^ mode is mainly related to the vibration of cation atoms. If we considered the mass increase due to Mg substituting Zn, we can expect a blue shift of the E_2_^Low^ mode, as was reported previously [34]. Generally, the E_2_^Low^ mode is less sensitive to induced pressure than the E_2_^High^ mode [32], thus a shift in case of the E_2_^Low^ mode is expected to be lower (Figure 10c).

### 3.5. XPS Study

The XPS method was used to confirm and identify the chemical character of Sb atoms detected in the surface area of the Zn_1−x_Mg_x_O:Sb selected films. High resolution XPS spectra of Sb 3d states are presented in Figure 11. Due to spin-orbit coupling, the Sb 3d state is split into two components (3d_5/2_ and 3d_3/2_). Unfortunately, the Sb 3d_5/2_ spectra are partially overlapped by O 1s peaks. Because the Sb 3d_3/2_ component is not affected by the oxygen signal, this peak can be used for the analysis. Deconvolution of the Sb 3d_3/2_ spectra reveals two chemical states of Sb. The peaks at about 539.7 eV are identified as Sb^3+^ oxidation states [35,36]. The second peak, located at 538.0 eV (Figure 11), can be assigned to the metallic Sb^0^ state [37] or alternatively to negatively charged Sb^3−^ or Sb^2−^ states. Tang et al. identified the XPS peaks related to 3d_5/2_ states of Sb^2−^ and Sb^3−^ states at 527.6 and 526.9 eV, respectively. Both chemical states were also evidenced in Zn_4_Sb_3_ materials. In Zn_1−x_Mg_x_O:Sb, the 3^−^ charge state is expected for Sb ions occupying oxygen sites [37]. In our alloy sample the contribution of the 5^+^ state is not observed, as it was reported earlier in case of pure ZnO:Sb layers [30]. Moreover, the magnesium content does not significantly influence the Sb chemical state.

### 3.6. Surface Morphology

Although the surface roughness does not influence the performance of photodetectors, high quality films with good morphology are crucial for light emitters. It is interesting to investigate how the Mg concentration affects the roughness parameters in polar and non-polar layers. The surface morphology of the deposited films was observed by atomic force microscopy (AFM) and the results are shown in Figure 12. A strong influence of the substrate orientation and Mg concentration on the crystalline growth can be clearly observed. The root mean square (RMS) roughness evaluated from AFM images from the 2 × 2 µm region increases due to Mg content in nonpolar layers and the opposite behavior is observed in the case of polar oxide layers (0001-oriented) grown both on *a*- and *c*-sapphire (Figure 13). The smallest *rms* is observed for the samples grown on *r*-oriented substrates.

## 4. Summary and Conclusions

Sb-doped Zn_1−x_Mg_x_O layers with various concentration of Mg were successfully grown by plasma-assisted MBE on differently oriented Al_2_O_3_ substrates. The presence of the Sb dopant was confirmed by the SIMS measurements, which showed a homogenous distribution of Sb at the level of 2 × 10^20^ cm^−3^, independent of the Mg content. X-ray diffraction confirmed the polar orientation of the layers grown on the *c*-and *a*-oriented sapphire and non-polar orientation on the *r*-oriented substrate. A comparison of the FWHM of the 0002 XRD reflexes for layers grown on polar Al_2_O_3_ substrates exhibits lower crystallographic quality for c orientation than for a. An increase in Mg content decreases the crystallographic quality of the layers and leads to the appearance of low intensity peaks correlated with other wurtzite orientations. The orientation of the substrates also affects the surface morphology, and the smoothest surface is observed in the case of layers grown on *r*-oriented sapphire. The *rms* parameter increases with increasing Mg content in all nonpolar samples and decreases in polar ones.

Raman modes related to Sb are visible in all doped samples. It can be noted that the spectra of *c*- Zn_1−x_Mg_x_O:Sb/*a*-Al_2_O_3_ samples exhibit most of the features associated with the Sb doping: the 500–600 cm^−1^ band, as well as the 240 cm^−1^ line. The increase in Mg content affects the crystal symmetry and, as a consequence, the FWHM of the Raman ZnO modes increases, and therefore, for the layers with the highest Mg content, the E2 modes are undetected. Based on the E_2_ peaks positions, a strain analysis was carried out. It can be noted that all samples reveal a compressive strain that was greater for the *a*-oriented layers than for the *c*-oriented ones. It has also been found that the addition of Mg introduces a tensile strain along the xy plane that counteracts the compressive strain originating from the Sb dopant and the sapphire substrate [26]. XPS analysis shows that the Sb atoms are in their 3+ charge state, which may favor the formation of Sb_Zn_-2V_Zn_ acceptor complexes. We do not observe the influence of Mg atoms on the chemical state of Sb.

Mg doping also influences optical parameters of the layers. The increase in the FWHM values of the PL peaks with an increasing Mg content probably results from fluctuations in the alloy composition. Compared to other wide band gap materials, e.g., III-V materials, the zinc oxide-based alloys exhibit larger broadening parameters. The exciton Bohr radius in Zn_1−x_Mg_x_O is relatively small and therefore it may be more affected by a local fluctuation of Mg content. In samples with the highest concentration of Mg, the PL peaks overlap, and this effect can also contribute to the “s” shaped peak behavior in temperature dependent PL. Despite identical growth conditions, the PL peaks’ positions depend on the orientation of the sapphire substrates, and the differences in peak position increase with the Mg content.

The band gap energies extracted from the transmittance differ slightly for *a*, *c,* and *r* orientations and these differences increase with the increase of the Mg content. The differences between the energy gaps and the PL peak positions are strictly correlated with the Stokes shift, which increases with the increase in Mg content in Zn_1−x_Mg_x_O alloys. This shift depends on the substrate used.

## Figures and Tables

**Figure 1 materials-15-08409-f001:**
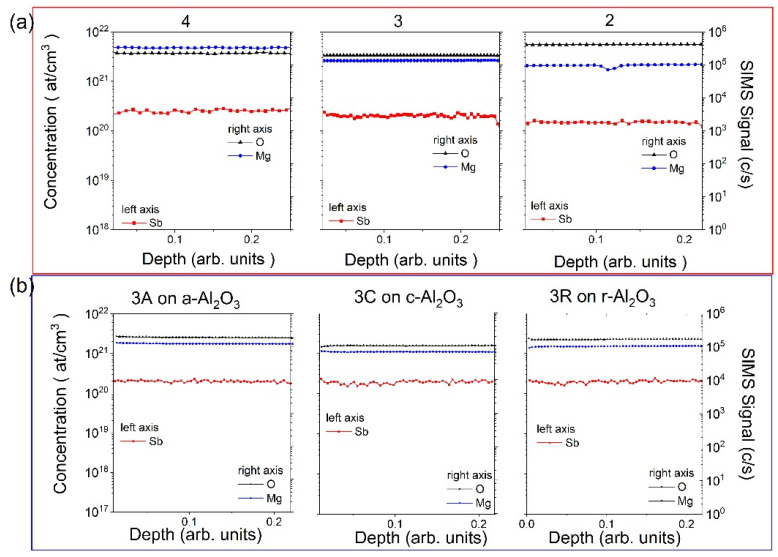
SIMS depth profiles for (**a**) a series of Zn_1−x_Mg_x_O samples with different concentration of Mg. Blue line corresponds to the Mg signal (right axis), red line refers to the Sb concentration obtained based on ion implantation standards (left axis); (**b**) a series of samples with the same concentration of Mg but grown on differently oriented sapphire substrates.

**Figure 2 materials-15-08409-f002:**
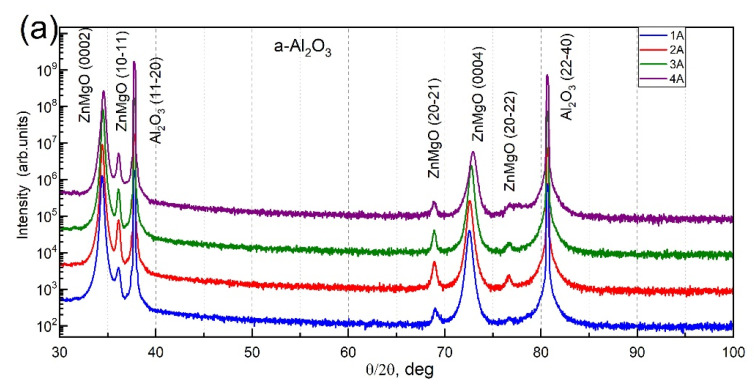
Theta–2Theta XRD patterns of the Zn_1−x_Mg_x_O:Sb layers (**a**) on *a*-Al_2_O_3_ (**b**) on *c*-Al_2_O_3_ (**c**) on *r*-Al_2_O_3_.

**Figure 3 materials-15-08409-f003:**
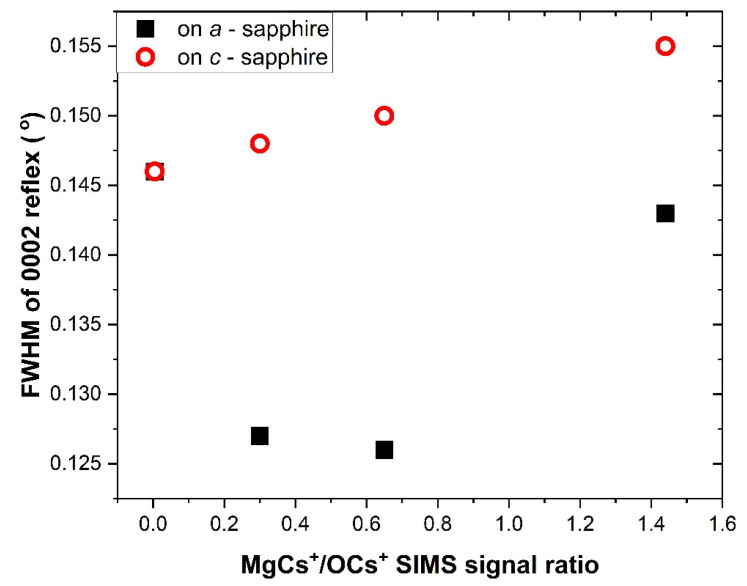
Full width a half maximum of 0002 peaks in the samples grown on *a*-sapphire (black squares) and on *c*-sapphire (red circles).

**Figure 4 materials-15-08409-f004:**
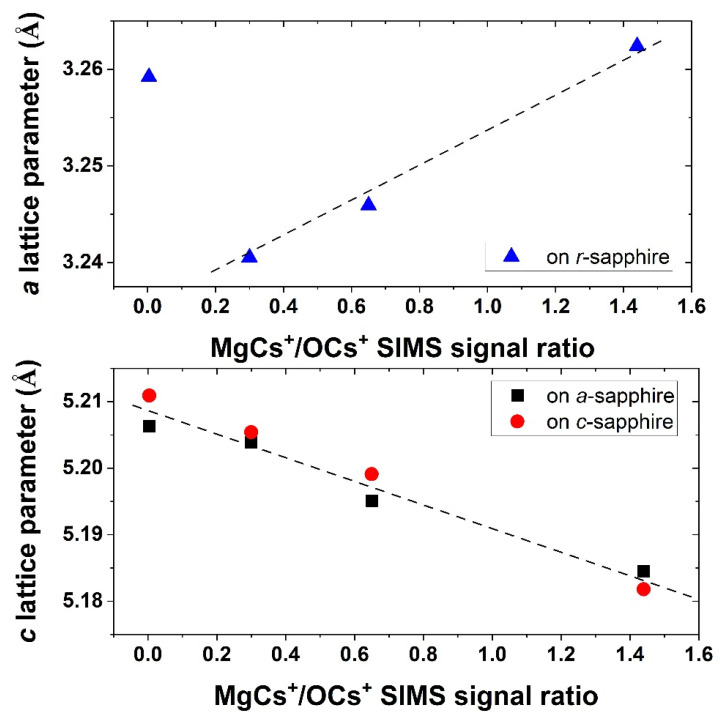
The *a*- and *c*-lattice parameters of Zn_1−x_Mg_x_O layers determined from XRD for nonpolar and polar layers, respectively, as a function of MgCs+/OCs+ SIMS signal ratio.

**Figure 5 materials-15-08409-f005:**
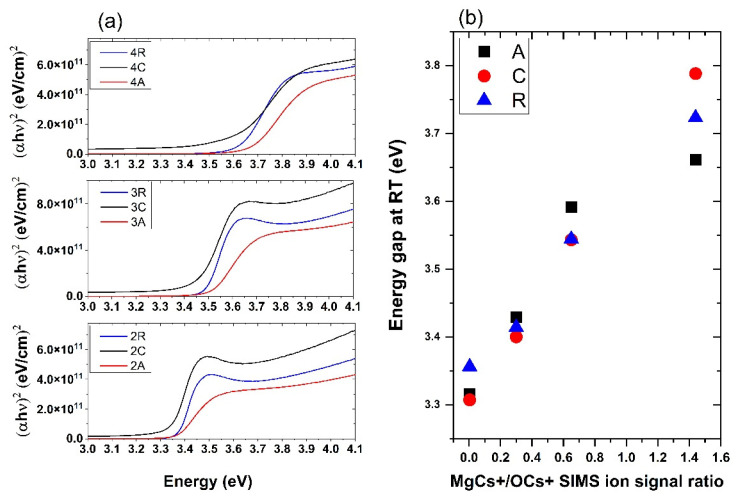
(**a**) Tauc plots for series of Zn_1−x_Mg_x_O samples grown on differently oriented sapphire substrates. (**b**) Energy gap vs. MgCs^+^/OCs^+^ SIMS ion signal ratio.

**Figure 6 materials-15-08409-f006:**
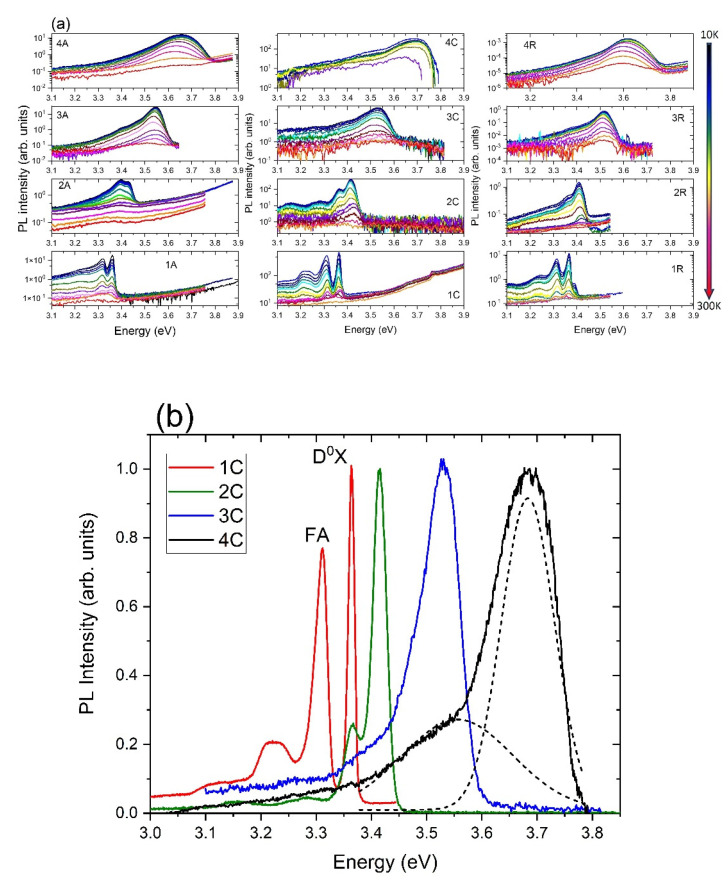
(**a**) Temperature dependence (10–300 K) of the Zn_1−x_Mg_x_O layers grown on *a*-, *c*- and *r*-sapphire band edge luminescence in a semi log scale. (**b**) Low temperature (10 K) normalized PL spectra of the samples 1C–4C.

**Figure 7 materials-15-08409-f007:**
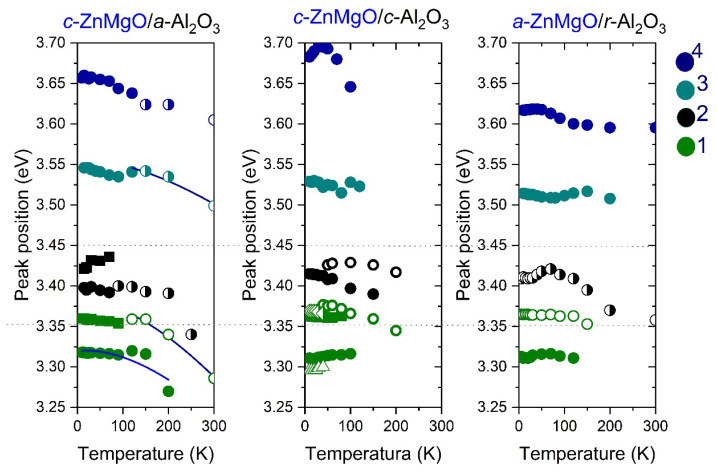
Analysis of the PL peaks position in series of Zn_1−x_Mg_x_O thin films. The same symbol colors correspond to the position of different PL peaks in the samples with intentionally the same Mg content.

**Figure 8 materials-15-08409-f008:**
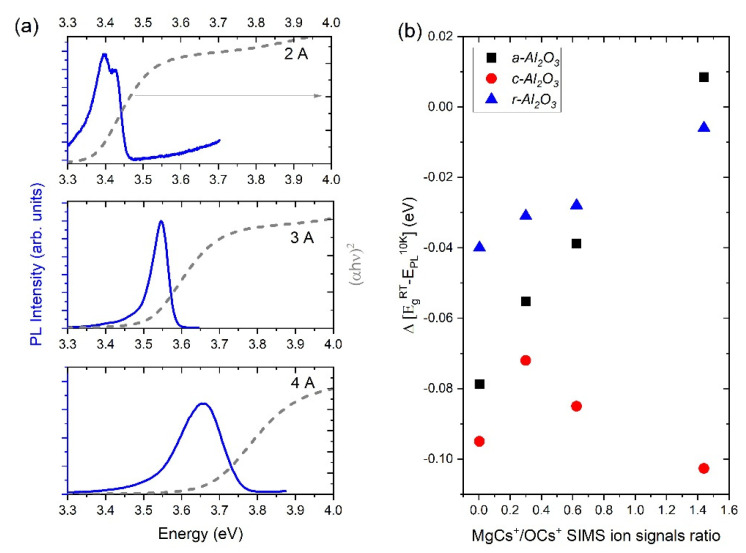
(**a**) PL measured at 10K (blue continuous line) and absorption measured at RT (grey line) extracted from transmittance of the Zn_1−x_Mg_x_O epitaxial films. (**b**) Delta parameters strictly related to Stokes shift as a function of the MgCs+/OCs+ ion signals measured by SIMS.

**Figure 9 materials-15-08409-f009:**
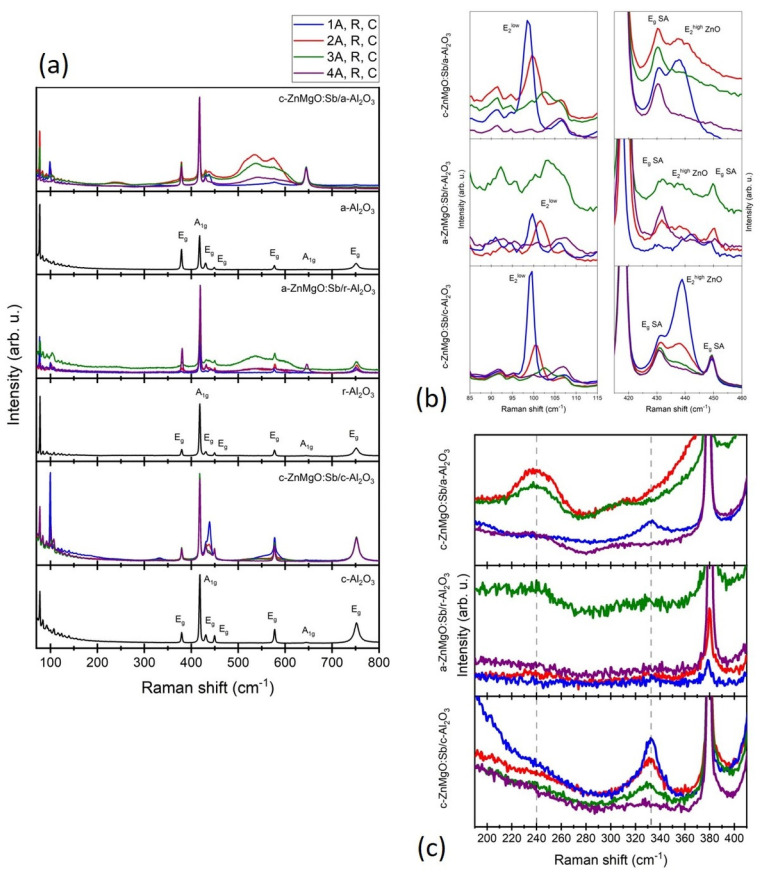
(**a**) Room temperature Raman spectra of *c*-Zn_1−x_Mg_x_O:Sb/*a*-Al_2_O_3_, *a*-Zn_1−x_Mg_x_O:Sb/*r*-Al_2_O_3_, and *c*-Zn_1−x_Mg_x_O:Sb/*c*-Al_2_O_3_ samples with corresponding substrates; λ_exct_ = 514.5 nm, P_exct_ = 20 mW; (**b**) enlarged part of Raman spectra showing ZnO E_2_ modes–SA corresponds to Al_2_O_3_ (sapphire) modes, while ZnO are the zinc oxide modes; (**c**) enlarged part of Raman spectra between 190 and 410 cm^−1^, showing the presence of 240 cm^−1^ and 333 cm^−1^ lines (marked with grey dashed line) in some spectra.

**Figure 10 materials-15-08409-f010:**
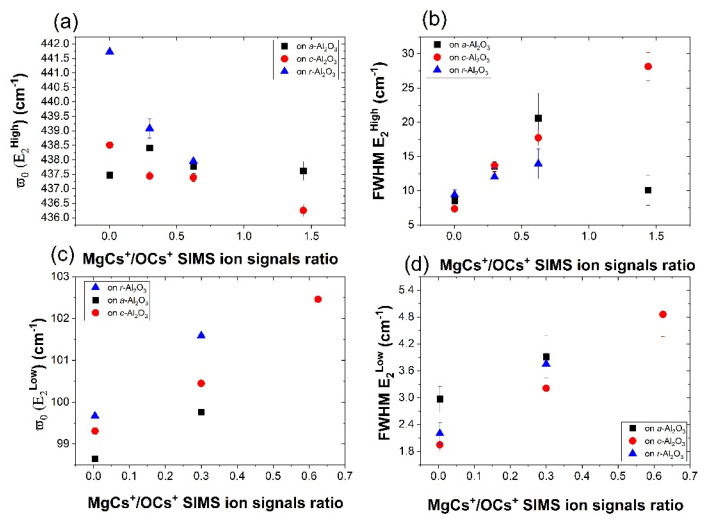
(**a**,**c**) Phonon frequency ω_o_ and (**b**,**d**) with the Raman E_2_^high^ and E_2_^Low^ mode as functions of Mg/O content in Zn_1−x_Mg_x_O.

**Figure 11 materials-15-08409-f011:**
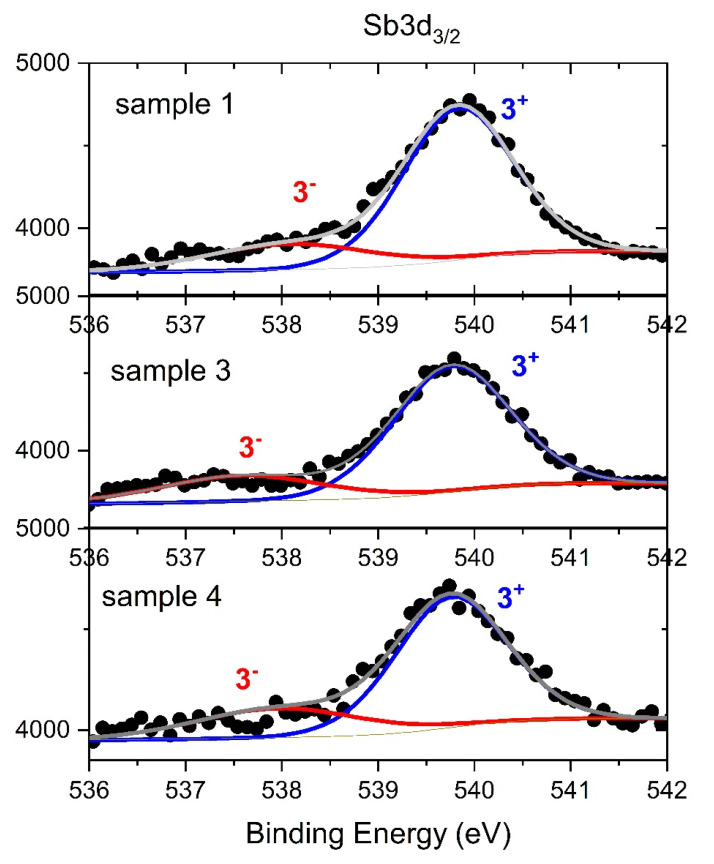
Sb 3d3/2 XPS spectra of ZnO films. Black points-XPS data, solid lines-analysis.

**Figure 12 materials-15-08409-f012:**
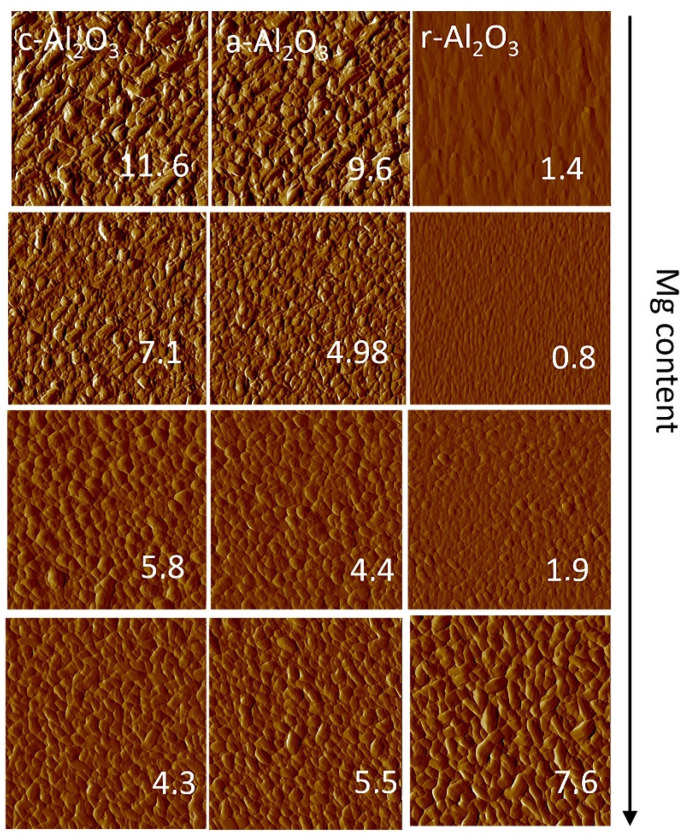
2D AFM images of Zn_1−x_Mg_x_O thin films.

**Figure 13 materials-15-08409-f013:**
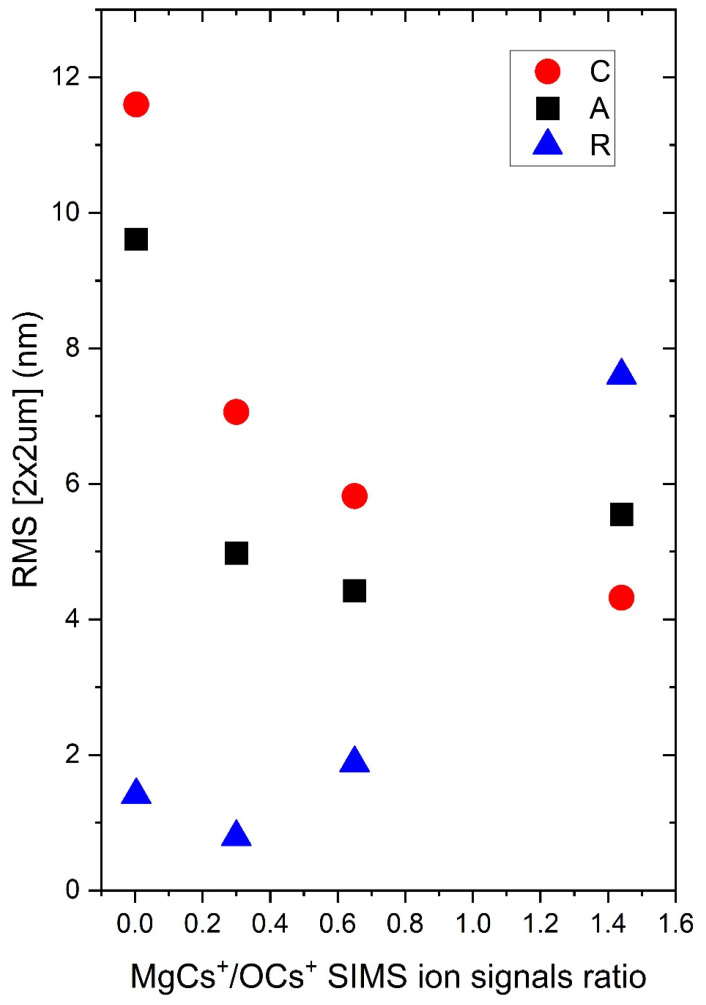
Root mean squares measured from 2 × 2 µm area on samples grown on *c*-, *r*-, *a*-oriented sapphires substrates.

**Table 1 materials-15-08409-t001:** Samples of Zn_1−x_Mg_x_O:Sb layers. NA-Non applicable.

Sample	Mg_Tip/Base_ Temperatures (°C)	Sb_Tip/Base_ Temperatures (°C)
1	NA	NA 460/300
2	555/385	
3	592/422	
4	608/438	

**Table 2 materials-15-08409-t002:** Energy gaps (Eg) obtained at RT.

Sample	MgCs+/OCs+ SIMS Signal Ratio	Eg on *a*-Sapphire (eV)	Eg on *c*-Sapphire (eV)	Eg on *r*-Sapphire (eV)
1	0.004	3.32	3.31	3.36
2	0.3	3.43	3.40	3.41
3	0.65	3.59	3.54	3.54
4	1.44	3.66	3.79	3.72

**Table 3 materials-15-08409-t003:** Summary of data for *c*-Zn_1−x_Mg_x_O:Sb/*a*-Al_2_O_3_, *a*-Zn_1−x_Mg_x_O:Sb/*r*-Al_2_O_3,_ and *c*-Zn_1−x_Mg_x_O:Sb/*c*-Al_2_O_3_ layers, containing Mg concentrations, positions, and FWHM of Raman E_2_ modes, obtained from fitting data with Lorentz functions.

MgCs+/OCs+ SIMS Ion Signals Ratio	Sample Name	ω (E_2_^low^)	FWHM (E_2_^low^)	ω (E_2_^high^)	FWHM (E_2_^high^)
		cm^−1^	cm^−1^	cm^−1^	cm^−1^
	*c*-Zn_1−x_Mg_x_O:Sb/*a*-Al_2_O_3_				
0.3	2A	99.76 ± 0.05	3.91 ± 0.47	438.41 ± 0.09	13.54 ± 0.70
0.004	1A	98.65 ± 0.04	2.97 ± 0.29	437.47 ± 0.14	8.55 ± 0.81
0.625	3A	-	-	437.77 ± 0.23	20.59 ± 3.69
1.44	4A	-	-	437.62 ± 0.32	10.09 ± 2.20
	*a*-Zn_1−x_Mg_x_O:Sb/*r*-Al_2_O_3_				
0.3	2R	101.59 ± 0.04	3.75 ± 0.26	439.08 ± 0.33	12.02 ± 1.29
0.004	1R	99.67 ± 0.04	2.20 ± 0.24	441.73 ± 0.12	9.41 ± 0.71
0.625	3R	-	-	437.95 ± 0.54	13.94 ± 2.17
1.44	4R	-	-	-	-
	*c*-Zn_1−x_Mg_x_O:Sb/*c*-Al_2_O_3_				
0.3	2C	100.45 ± 0.03	3.21 ± 0.23	437.44 ± 0.16	13.69 ± 0.95
0.004	1C	99.31 ± 0.02	1.95 ± 0.14	438.51 ± 0.07	7.37 ± 0.39
11.2	3C	102.46 ± 0.06	4.86 ± 0.49	437.39 ± 0.14	17.73 ± 1.13
1.44	4C	-	-	436.25 ± 0.20	28.15 ± 2.06

## Data Availability

The data presented in this study are available on request from the corresponding author.

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
