# Peer review of "Polar and Non-Polar Zn1−xMgxO:Sb Grown by MBE"

_materials, 2022, doi:10.3390/ma15238409_

Round 1

Author Response

­Reviewer 1

  • Page 1, line 44. More introduction/background should be provided. For example, why is Sb studied. Some key findings of this work can be highlighted at the end of introduction.

The introductory part has been rewritten and we hope that the current version provides a better justification for studies of Sb. 

  • Page 3, Table 1. What is the meaning of NA? where are A, C, R (page 2, line 89, “letters A, C and R next to the sample numbers”)?

NA-means "Not applicable"- an appropriate comment is added to the text. The designations A, C, R are used in the figures and in text to denote the orientation of the sapphire substrate.

  • Page 3, Figure 1(a). It seems the Sb (red) also decreases from 4M to 2M. By the way, what is “M” in the subfigure title?

Thank you for this remark, the letter M was used by mistake and Figure 1 has been corrected.

  • Page 4, Figure 2. What is the reason of ordering in 2, 1, 3 ,4.

The ordering has  been corrected.

  • Why does only 4C show obvious (20-22)? There is no discussion.

For sample 4C, with the highest concentration of Mg, X-rays reflected from the second plane of atoms (20-22) are observed. In this sample the intensity of X-rays reflected from the first planes of atoms (10-11) is strong what makes observation of the second planes possible. The intensity of (20-22) reflection is about 20 times smaller than that of (10-11) (according to the peaks intensity relationship in fully polycrystalline ZnO powders). Therefore, the (20-22) reflexes are not detected in samples with weak (10-11) signals. 

  • (c) why does 2R sample have greater noise?

Thank you to the reviewer for pointing out, this has been corrected. The greater noise was to some extent an “illusion” associated with the use of the semi-log scale.

  • Page 4, line 131. There is no (0001) peak marked in (c)?

The (0001) reflection is forbidden for ZnO wurtzite structure. The “(0001)” used in the text indicate crystallographic orientation of a ZnMgO layer.

  • Page 5, line 142. Where does the “high Mg concentration exhibit polycrystalline properties” come from.

As explained in the text page 8 “The problem with the homogeneity of Zn1−xMgxO for higher Mg concentration is repeatedly reported and is correlated with the fact that MgO crystallizes in a cubic structure instead of a hexagonal one.13,14 The thermodynamic solubility limit of MgO in the wurtzite phase of Zn1−xMgxO alloy was reported to be x = 0.04 in a bulk form 15 and the solubility limit is much higher for epitaxial thin films. In our samples with a high concentration of Mg pure wurtzite phases were detected but with polycrystalline character and with a preferential orientation.”

  • Page 6, section 3.2. Page 7, Figure 5. From reading this, it is still hard to know what exact bandgap of each sample has. The author should use a table with precise values to summarize them.

The band gap energy values are presented now in Table 2 and in Fig 5 (b).

  • Page 8, line 182-194. The discussion here is very messy and hard to follow. The author should reorganize it and emphasize key points.

The discussion presented on page 8 has been rewritten to be more precise.

  • Page 9, line 241. Why is the Sb doping related to Mg concentration in Figure 9(a)(1)?

The Reviewer did not explain what 9(a)(1) means – whether it is all samples 1 shown in Figure 9a or first panel of Figure 9a. In the first case, it is worth noting that samples marked with number 1 do not contain any antimony, these are pure ZnO samples, therefore Raman spectra of these samples do not contain 500-600 cm-1 band related to the presence of Sb. In the second case, such tendency is observed in all Zn1-xMgxO:Sb samples, however, in the case of c-ZnMgO:Sb/a-Al2O3 samples the intensity of the signal from this band is the largest so this dependency is easily visible.

Generally, the presence of MgO in the ZnO lattice may influence how dopants incorporate into the crystal lattice. Such a dependence was shown for ZnMgO layers doped with arsenic [E. Przezdziecka et al. Applied Surface Science, vol. 435, pp. 676–679, 2018], which is another group V element used for obtaining p-type conductivity in ZnO. Studies showed that increasing Mg concentration affects the relative proportions of As-related defects and for higher Mg content the contribution of acceptor complex AsZn −2VZn defect increased from around 50% for 4% Mg sample up to 70% for 12.3% Mg sample, simultaneously suppressing contribution of AsZn defect down to 20%, indicating that doping of ZnMgO alloys with group V elements can be favorable for obtaining p-type conductivity. However, this aspect of our studies was not discussed in the reviewed article because it is a quite complex topic and the article seemed already quite long. We decided to leave this part for another article to be prepared.

  • Page 10, Figure 9(b). Why do E2 high ZnO of sample 1A and 1C looks very different?

The intensity of the Raman signal for samples grown on c-Al2O3 was larger than for samples grown on a-Al2O3, therefore the spectra for C samples seem to be smoother. However, in both cases the ZnO E­2high mode in very well distinguishable.

  • Page 11, Figure 10. o How is the error bar obtained? Why do some samples have error bar, others not?

The values of error bars for each data point in Figure 10 are included in Table 3. Their values come from fitting of every Raman line with Lorentz function. Each point has error bars, but sometimes they are so small that they are not visible due to the size of the data point.

  • Line 311-313. What is the reason of this phenomenon? o FWHM is provided. But there is no analysis on it.

The phenomenon is explained in text. Mode E2high shifts more for samples grown on r-Al2O3 because they are more strained than layers grown on a- and c-Al2O3. General analysis on FWHM is provided in lines 264-291. More complex analysis will be included in another paper regarding these samples that is in writing.

  • Page 13, line 356. But when Mg concentration is high, the relationship reverses? Any explanation?

In page 13 XPS data was disused and we observed that “the magnesium content does not influence significantly Sb chemical state”

  • Page 15. The conclusion part could be more concise.

The conclusion part has been revised.

Minor Improvement:

  • Page 3, line 102. After SIMS test, the author should conclude what ‘x’ exactly is in Zn1−xMgxO. After reading the whole paper, the reviewer still can’t tell how much ‘x’ is.

To determine dopant concentration by SIMS method, the appropriate standard is necessary. Usually, it is the proper material e.g ZnO implanted with suitable element e.g. Mg. The calculation of concentration of element is then proceeded by using RSF (Relative Sensitivity Factor) as calculated from the SIMS signals of measured standard.

Unfortunately, when the concentration of element increase to the amount which makes it the matrix element (like in MgxZn1-xO), the so called “matrix effect” preclude using the RSF. To obtain the exact content of ternary compound the calibration curve is needed as described [1].

[1] DOI: 10.1002/sia.6705

  • Page 5, Figure 3. o Title. “with” à “width” Page 7 line 181. Higher than or lower than?

It was corrected

  • Page 8, line 185. Where are the two peaks of sample 4?

For better visualization, the deconvolution of the PL spectra for sample 4 was added to Figure 6.

  • Page 9, line 206. The Stoke’s shift should be defined.

Approproate comment was added to the text. “Stokes shift is the difference between positions of the absorption and emission spectra of the same electronic transition.”

Reviewer 2 Report

In this paper, the authors demonstrated the fabrication of Zn1-xMgxO: Sb-doped with different magnesium contents by molecular beam epitaxy on different oriented sapphire substrates.  The presence of Sb atoms was confirmed and analyzed by SIMS, XPS, and Raman spectroscopy. Moreover, the optical and structural parameters were investigated by luminescence and transmittance as well as XRD, respectively. The paper provides interesting results and can direct to more advancements in the field; however, it needs some modifications before resubmission. Here are some comments:

1.     The abstract needs to be rewritten to give more comprehensive details about the work.

2.     The introduction should give a state-of-the-art for similar published papers and the advantages and disadvantages of the used methods and techniques compared to the presented ones. In this regard, the novelty of the work should be highlighted.

3.     Quantitative values of the bandgaps should be given

4.     The results presented in Fig. 7 and Fig. 8 need to be declared and need more discussion (Note that all figures should be cited in the manuscript; for instance: Fig. 8(a) and Fig. 8(b) should be discussed and highlighted separately)

5.     More quantitative results are needed to be included in the conclusion

6.     It will be advantageous if the authors measure the electrical properties (hole concentration and mobility) of Zn1−xMgxO:Sb films by Hall measurements for different x. it will be interesting to show that in ZnO:Sb films, only n-type conduction is achieved, whereas p-type conduction is accomplished in other x-compositions.  This study will extremely enrich the work.

7.     The overall writing skill of this paper is not so good. There are many grammatical mistakes, which must be pointed out by the authors and corrected subsequently.

Author Response

Revivers 2

  • The abstract needs to be rewritten to give more comprehensive details about the work.

New abstract was prepared

  • The introduction should give a state-of-the-art for similar published papers and the advantages and disadvantages of the used methods and techniques compared to the presented ones. In this regard, the novelty of the work should be highlighted.

The introduction part is improved.

  • Quantitative values of the bandgaps should be given

The values of Eg was presented now on Fig 5 (b) and in Table 2.

  • The results presented in Fig. 7 and Fig. 8 need to be declared and need more discussion (Note that all figures should be cited in the manuscript; for instance: Fig. 8(a) and Fig. 8(b) should be discussed and highlighted separately)

Appropriate corrections have been made in the text.

  • More quantitative results are needed to be included in the conclusion

The conclusion part is improved.

  • It will be advantageous if the authors measure the electrical properties (hole concentration and mobility) of Zn1−xMgxO:Sb films by Hall measurements for different x. it will be interesting to show that in ZnO:Sb films, only n-type conduction is achieved, whereas p-type conduction is accomplished in other x-compositions.  This study will extremely enrich the work.

We thank the reviewer for this suggestion, we plan to perform such measurements in the future. Due to problems encountered with the quality of ohmic contacts, we decided not to include incomplete and inaccurate electrical results in this paper.

  • The overall writing skill of this paper is not so good. There are many grammatical mistakes, which must be pointed out by the authors and corrected subsequently.

The proofreading of the work has been done.

Round 2

Reviewer 1 Report

Thanks the author for addressing all the comments and making revision. 

A minor point:

The order/color of lines in Figure 2(c) are still different from (a)(b). No sure if this is a mistake or intentional.

Author Response

Thank you for this reviewer's note, it has been corrected.

Reviewer 2 Report

The authors have taken the raised issues into consideration. The manuscript is now improved; so, I recommend that the paper be accepted for publication.

Author Response

We are grateful for any comments that have helped to improve this manuscript. We are glad that the current version is ready for publication.